# Detecting Floral Resource Availability Using Small Unmanned Aircraft Systems

Nicholas V. Anderson [1], Steven L. Petersen [1], Robert L. Johnson [2], Tyson J. Terry [3] and Val J. Anderson [1,*]

1. Department of Plant and Wildlife Sciences, Brigham Young University, Provo, UT 84602, USA; nva22@byu.edu (N.V.A.); steven_petersen@byu.edu (S.L.P.)
2. Department of Biology, Brigham Young University, Provo, UT 84602, USA; robert_johnson@byu.edu
3. Disturbance Ecology Department, University of Bayreuth, 95444 Bayreuth, Germany; tyson.terry@uni-bayreuth.de
* Correspondence: val_anderson@byu.edu

**Abstract:** Floral resources for native pollinators that live in wildland settings are diverse and vary across and within growing seasons. Understanding floral resource dynamics and management is becoming increasingly important as honeybee farms seek public land for summer pasture. Small Unmanned Aircraft Systems (sUASs) present a viable approach for accurate broad floristic surveys and present an additional solution to more traditional alternative methods of vegetation assessment. This methodology was designed as a simplified approach using tools frequently available to land managers. The images of three subalpine meadows were captured from a DJI Phantom 4 Pro drone platform three times over the growing season in 2019 in Sanpete County, Utah. The images were composited using Pix4D software 4.5.6 and classified using a simple supervised approach in ENVI 4.8 and ArcGIS Pro 2.4.3 These same meadows were assessed using two traditional ocular methods of vegetation cover–meter-squared quadrats and macroplot estimation. The areas assessed with these methods were compared side by side with their classified counterparts from drone imagery. Classified images were not only found to be highly accurate when detecting overall floral cover and floral color groups (76–100%), but they were also strongly correlated with quadrat estimations, suggesting that these methods used in tandem may be a conducive strategy toward increased accuracy and efficiency when determining floral cover at broad spatial scales.

**Keywords:** drone technology; remote sensing; floral resource detection; vegetation mapping; pollinator resources

## 1. Introduction

Monitoring seasonally available resources is an important aspect of land management. Land cover surveys and vegetation measurements are used by land managers as indicators of seasonal resource dynamics and availability. There are many traditional methods used for measuring vegetation cover such as line point, line intercept, quadrat estimation, point-frame, the ocular macroplot, and others [1–3]. When monitoring areas across broad temporal or spatial scales, objective methods, such as line point or point frame methods can become both time and labor-intensive and provide cover estimates that typically represent a very limited percentage of the total sample area. While traditional ocular methods can be easily and rapidly employed by an individual or field crew, the unique perception of each individual technician collecting field data can result in variable and even biased results. Remote sensing techniques, however, present an objective, efficient, landscape-scale solution with high-accuracy cover estimations [4].

The aerial imagery acquired from satellites has been widely utilized to classify terrestrial ecosystems and to monitor temporal changes in vegetation structure. In some instances, individual plant species can be identified by their particular reflective properties (recorded as spectral signatures) from space-borne sensors [5]. However, many of the





seasonal resources important to land managers, such as grass, forbs, and shrubs, are too small and variable to be accurately monitored by satellites due to the coarse scale of the imagery and the low frequency at which it is collected. Large, manned aircraft can be used to obtain broad-scale, high-resolution imagery for measuring vegetation. For example, Hulet et al. [6] accurately detected individual pinyon pines (*Pinus edulis*) and Utah junipers (*Juniperus osteosperma*), and Petersen et al. [7] identified individual willow species within a northwestern riparian area using high-resolution imagery collected from manned aircraft. Unfortunately, the use of large numbers of manned aircraft to obtain imagery can be difficult to schedule and expensive, particularly when frequent monitoring is needed.

With a much finer spatial resolution compared to typical satellite or aircraft-based sensors, small, Unmanned Aircraft Systems (sUASs) present a newer and potentially more efficient method of surveying vegetative resources within and across seasons, which greatly increases the detection of spatiotemporal resources and dynamic shifts at very fine scales [8]. High-resolution sUAS imagery has been effectively used for accurately mapping rangeland vegetation, including species-level discrimination for several species [9]. Forest canopies have been imaged to help explain spatial patterns of biodiversity [10]. Snow depth in the Artic has been measured to help ecological and civil engineering research [11]. Coastal marine habitats have been analyzed to aid in mapping fish nursery areas [12]. The use of sUAS is spreading rapidly within different disciplines and has been promoted as a complementary tool to traditional field surveying methods [10].

Under the multiple-use management regime developed for public lands in the United States, the US Forest Service (USFS) and Bureau of Land Management (BLM) have incorporated the summer pasturing of commercial honeybee hives with other uses such as livestock grazing, recreation, energy development, and timber harvest [13]. An interesting association with the management and placement of beehives is the availability of nectar and pollen in the proximity of potential hive placement. When pasturing any species of non-native animals in natural settings, it is important for land managers to determine stocking rates that do not significantly impair native species' population dynamics. Practices to measure resource availability have been established to balance the resource use of livestock and native ungulates; however, no such practices have been established to measure resource availability for pollinators. In order to develop stocking rates for managed honeybees, a critical first step is the quantification of available pollinator resources, i.e., the pollen and nectar in flowering plants and the associated floral cover in a given area.

The objectives of this study were (1) to assess the efficacy of using high-resolution imagery from sUAS to accurately calculate floral cover and (2) to compare the results of classified aerial imagery between two traditional methods of vegetation cover analysis: quadrat sampling and ocular macroplot estimation. In contrast to the hyperspectral and deep learning methods recently presented by Gallmann et al. [14] and Barnsley et al. [15], this study utilizes a standard off-the-shelf drone platform and RGB sensor to assess floral cover providing an affordable, simplistic, and rapid solution for technicians and land managers currently utilizing traditional methods.

## 2. Materials and Methods

Three subalpine meadows in Ephraim Canyon, Sanpete County, UT, USA, were selected as the study locations. At each location, a 0.5-hectare macroplot was delineated using a 40 m rope stretched from a central stake. Aerial imagery was collected using a DJI Phantom 4 quadcopter equipped with a standard 20 mm red, green, and blue (RGB) sensor at 10 m above ground level (AGL). Flights were automated through the Pix4D Mapper app in a grid-like fashion with 70% overlap. Flights over the study locations occurred three times throughout the flowering season on 7 July 2019, 6 August, and 20 August. Aerial images were organized and stitched together using Pix4D software 4.5.6 (https://www.pix4d.com/product/pix4dmapper-photogrammetry-software) to create a single high-resolution orthomosaic of each study site, resulting in a total of nine distinct orthomosaics. The software program ENVI 4.8 (Exelis Visual Information Solutions, Boulder, CO, USA)

was employed to execute pixel-based supervised classification using a maximum likelihood classifier. Training data were collected for eight different cover classes: dark vegetation, light vegetation, bare ground, shadows, rocks, blue flowers, yellow flowers, and white flowers. These classes were selected after various test trials to determine (1) the classes that needed to be included to properly extract the three floral color classes and (2) the proper amount of variation that needed to be captured for each class. At least 30 training data samples were collected for each class, and more, if necessary, to encompass the variation within each class. This included training samples from each drone image for as many floral species as could be found within each color class. Research materials within the orthomosiacs were manually classified and removed from the analysis. Prevalent species varied both temporally and by site. A complete inventory of vegetation was conducted prior to this study at each site. These are included in Tables 1–3, along with their corresponding color categories.

Two traditional vegetation measurement methods were also implemented at each study site as follows: an ocular quadrat estimation and an ocular macroplot estimation (Figures 1 and 2). All cover estimates were collected by a single trained individual to reduce inter-technician bias. The ocular quadrat estimation consisted of ten one-meter squared quadrats that were randomly placed within each study area. Within these quadrats, floral cover was visually estimated for each flowering species and assigned their corresponding color classes. Each quadrat within the classified orthomosaics was then clipped for direct comparison with ocular quadrat estimates.

To perform the ocular macroplot estimation, a 40 m rope was used to walk around the circumference of the macroplot and visually assess the floral cover within. A fenced area was constructed within each macroplot to represent the 1% area of the macroplot as a visual reference when allocating cover percentages. Cover percentages from traditional methods and those generated from the classified drone images were compared in R software 4.3.1 (RStudio Team (2020) RStudio: Integrated Development for R. RStudio, PBC, Boston, MA, USA, URL http://www.rstudio.com/) using a Wilcoxon Rank Sum Test.

**Table 1.** Plant species located at Philadelphia Flat.

| Plant Species | Classified Category |
| --- | --- |
| *Artemesia ludoviciana* | Vegetation |
| *Bromus carinatus* | Vegetation |
| *Collomia linearis* | White |
| *Delphinium nuttallianum* | Blue |
| *Elymus trachycaulus* | Vegetation |
| *Erigeron speciosus* | Blue |
| *Erythronium grandiflorum* | Yellow |
| *Geranium viscosissimum* | White |
| *Ligusticum porteri* | White |
| *Lupinus argenteus* | Blue |
| *Melica bulbosa* | Vegetation |
| *Orthocarpus tolmiei* | Yellow |
| *Osmorhiza occidentalis* | White |
| *Penstemon rydbergii* | Blue |
| *Penstemon watsonii* | Blue |
| *Potentilla gracilis* | Yellow |
| *Stellaria jamesiana* | White |
| *Stipa lettermanii* | Vegetation |
| *Symphoricarpos oreophilus* | White |
| *Thalictrum fendleri* | Vegetation |
| *Vicia americana* | Blue |
| *Viguiera multiflora* | Yellow |
| *Viola purpurea* | Yellow |

**Table 2.** Plant species located at Ephraim Skyline.

| Plant Species | Classified Category |
| --- | --- |
| *Achillea millefolium* | White |
| *Aquilegia coerulea* | White |
| *Artemesia ludoviciana* | Vegetation |
| *Aster ascendens* | Blue |
| *Bromus carinatus* | Vegetation |
| *Castilleja rhexifolia* | Yellow |
| *Collomia linearis* | White |
| *Delphinium nuttallianum* | Blue |
| *Elymus trachycaulus* | Vegetation |
| *Erigeron flagellaris* | Blue |
| *Geranium viscosissimum* | White |
| *Lupinus argenteus* | Blue |
| *Melica bulbosa* | Vegetation |
| *Orthocarpus tolmiei* | Yellow |
| *Osmorhiza occidentalis* | White |
| *Penstemon rydbergii* | Blue |
| *Poa pratensis* | Vegetation |
| *Polemonium foliosissimum* | Blue |
| *Potentilla gracilis* | Yellow |
| *Senecio crassulus* | Yellow |
| *Stellaria jamesiana* | White |
| *Stipa lettermanii* | Vegetation |
| *Stipa nelsonii* | Vegetation |
| *Swertia radiata* | Vegetation |
| *Taraxicum officinale* | Yellow |
| *Thalictrum fendleri* | Vegetation |
| *Trisetum spicatum* | Vegetation |
| *Valeriana occidentalis* | White |
| *Vicia americana* | Blue |
| *Viola purpurea* | Yellow |

**Table 3.** Plant species located at Horseshoe Flat North.

| Species | Classified Category |
| --- | --- |
| *Achillea millefolium* | White |
| *Agoseris aurantiaca* | Yellow |
| *Artemisia ludoviciana* | Vegetation |
| *Astragalus tenellus* | Blue |
| *Bromus carinatus* | Vegetation |
| *Castilleja rhexifolia* | White |
| *Chrysothamnus viscidiflorus* | Yellow |
| *Collomia linearis* | White |
| *Delphinium nuttallianum* | Blue |
| *Elymus Trachycaulus* | Vegetation |
| *Erigeron speciosus* | Blue |
| *Erythronium grandiflorum* | Yellow |
| *Geranium viscosissimum* | White |
| *Madia glomerata* | Yellow |
| *Melica bulbosa* | Vegetation |
| *Orthocarpus tolmiei* | Yellow |
| *Penstamon rydbergii* | Blue |
| *Potentilla gracilis* | Yellow |
| *Stipa lettermanii* | Vegetation |
| *Stipa nelsonii* | Vegetation |
| *Vicia americana* | Blue |
| *Viola purpurea* | Yellow |

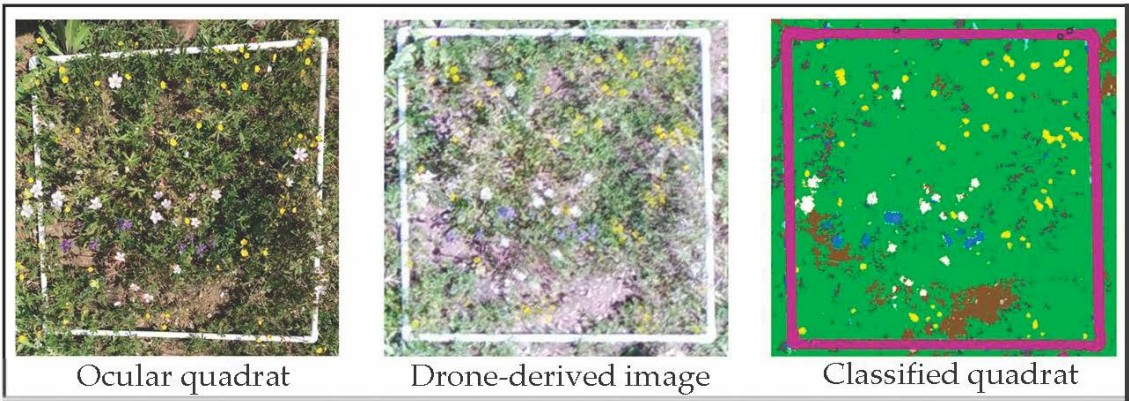

**Figure 1.** Example of classified and unclassified quadrats.

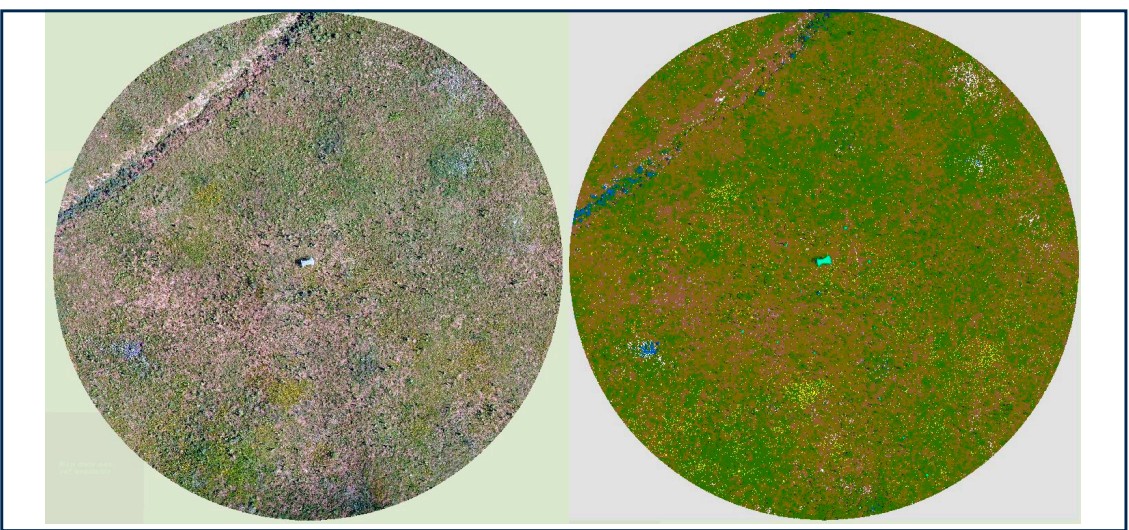

**Figure 2.** Example of unclassified (**left**) and classified (**right**) macroplot orthomosiacs.

An accuracy assessment of each classification was performed using the high-resolution base image as the reference. Fifty points were randomly selected within each cover class using a random points tool in ArcGIS Pro 2.4.3 (ESRI 2011. ArcGIS Desktop. Redlands, CA, USA: Environmental Systems Research Institute). Their accuracy was assessed and recorded within a confusion matrix to establish both the user's and producer's accuracy of each class and the overall accuracy for each classification. Additionally, Kappa statistics were produced for each classification.

## 3. Results

The overall accuracy for the nine classifications ranged between 76.9 and 88.0%, with an average accuracy of 83.0%. The overall accuracy of just the floral classes ranged between 74.0 and 100%, with an average accuracy of 86.7%. The accuracy of floral classification varied by color (Table 4). The average producer's accuracy for the blue flower class was 95.4%, for the yellow flower class was 97.4%, and for the white flower class was 92.5% (Table 5). The average user's accuracy for the blue flower class was 82%, for the yellow flower class was 90.22%, and for the white flower class was 86% (Table 5). A kappa statistic was generated to determine the strength of each classification. Kappa statistics ranged from 73.0 to 86.5%, with an average of 80.7% (Table 4).

**Table 4.** Summary statistics for the accuracy assessment of each classification.

| Study Site | Overall Accuracy | Overall Floral Accuracy | Kappa |
|---|---|---|---|
| Ephraim Skyline 7_23_19 | 0.77 | 0.81 | 0.74 |
| Ephraim Skyline 8_6_19 | 0.88 | 0.88 | 0.87 |
| Ephraim Skyline 8_20_19 | 0.82 | 0.85 | 0.80 |
| Horseshoe Flat 7_23_19 | 0.76 | 0.88 | 0.73 |
| Horseshoe Flat 8_6_19 | 0.85 | 0.88 | 0.83 |
| Horseshoe Flat 8_20 19 | 0.81 | 0.74 | 0.79 |
| Philly Flat 7_23_19 | 0.88 | 0.84 | 0.86 |
| Philly Flat 8_6_19 | 0.84 | 0.93 | 0.81 |
| Philly Flat 8_20_19 | 0.87 | 1.00 | 0.85 |
| Total Average | 0.83 | 0.87 | 0.81 |

**Table 5.** Accuracy of each color class for each classification.

| | Producers Accuracy | | | Users Accuracy | | |
|---|---|---|---|---|---|---|
| Study Site | Blue | Yellow | White | Blue | Yellow | White |
| Ephraim Skyline 7_23_19 | 93.88 | 94.12 | 95.56 | 92 | 64 | 86 |
| Ephraim Skyline 8_6_19 | 86.79 | 97.78 | 100 | 92 | 88 | 84 |
| Ephraim Skyline 8_20_19 | 97.37 | 98.04 | 93.02 | 74 | 100 | 80 |
| Horseshoe Flat 7_23_19 | 95.74 | 100 | 95.12 | 90 | 96 | 78 |
| Horseshoe Flat 8_6_19 | 93.62 | 98.00 | 75.00 | 88 | 98 | 78 |
| Horseshoe Flat 8_20 19 | 100 | 90.91 | 93.02 | 42 | 100 | 80 |
| Philly Flat 7_23_19 | 95.74 | 100 | 92.00 | 90 | 70 | 92 |
| Philly Flat 8_6_19 | 100 | 100 | 94.12 | 88 | 96 | 96 |
| Philly Flat 8_20_19 | - | 98.04 | 94.34 | - | 100 | 100 |
| Total Average | 95.39 | 97.43 | 92.46 | 82 | 90.22 | 86 |

Ocular measurements of the ten quadrats at each site were compared to the output estimates of the classified quadrats from the aerial images (Figure 3). The differences between quadrat measurements and their corresponding classifications are not normally distributed; therefore, a Wilcoxon Signed-Rank Test was used to observe this difference (Table 6). The average floral cover estimated using the classified imagery across all individual quadrats and floral colors was 1.21% versus 2.40% via ocular quadrat estimation. Overall, the floral cover between classified quadrat outputs and ocular quadrat estimates proved to be significantly different from one another (*p*-value 0.004), indicating from the accuracy of the drone-derived estimates that the ocular quadrat method over estimated floral cover by roughly two times. The same trend was reflected within each individual color class. The drone average estimates for each color class were blue at 0.42%, white at 0.27%, and yellow at 0.52%. By contrast, the quadrat averages for each color class were blue at 0.61%, white at 0.71%, and yellow at 1.08%.

Classified estimates and quadrant estimates were positively correlated between the floral classes and showed relatively strong relationships (Figure 4). Blue flowers had an r-squared value of 0.81, white flowers had a value of 0.80, and yellow flowers had a value of 0.66. The strength of the yellow class differed from the blue and white classes due to a single observation.

Similar to the quadrat evaluations, the macroplot evaluations were compared between ocular estimates and classified imagery (Figure 5). A Wilcoxon Signed-Rank Test was used to observe the differences (Table 7). The average floral cover estimated using the classified imagery over all macroplots and all floral colors was 0.9% versus 4.81% via ocular macroplot estimation. Overall, the floral cover between classified macroplot outputs and ocular macroplot estimates proved to be significantly different from one another (*p*-value 0.004). Like the ocular quadrat estimates, the ocular macroplot estimates over estimated floral cover by roughly five times. This trend held true across the three floral classes. The classified macroplot averages for each floral class were blue at 0.28%, white

at 0.23%, and yellow at 0.39%. In contrast, the ocular macroplot averages for each floral class were blue at 0.94%, white at 1.33%, and yellow at 2.33%, each differing significantly from the corresponding classified estimates. In both traditional methods, the ocular meter squared quadrat and the ocular macroplot, a significant overestimation of the floral cover was observed.

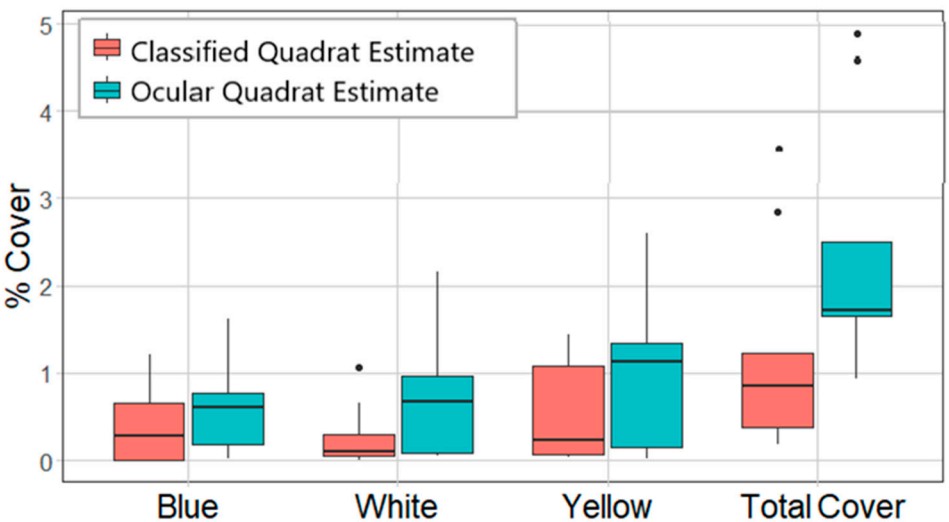

**Figure 3.** Comparison of floral cover between classified quadrat estimates and ocular quadrat estimates.

**Table 6.** Summary statistical results from the Wilcoxon Rank Sum Test comparing classified quadrat cover estimates to ocular quadrat cover estimates.

| Class | Classified Quadrat Average | Ocular Quadrat Average | Sig Diff | V Score | *p* Value |
|---|---|---|---|---|---|
| Blue | 0.42 | 0.61 | marginal | 6 | 0.055 |
| White | 0.27 | 0.71 | Yes | 2 | 0.0117 |
| Yellow | 0.52 | 1.08 | Yes | 5 | 0.039 |
| Total | 1.21 | 2.40 | Yes | 45 | 0.004 |

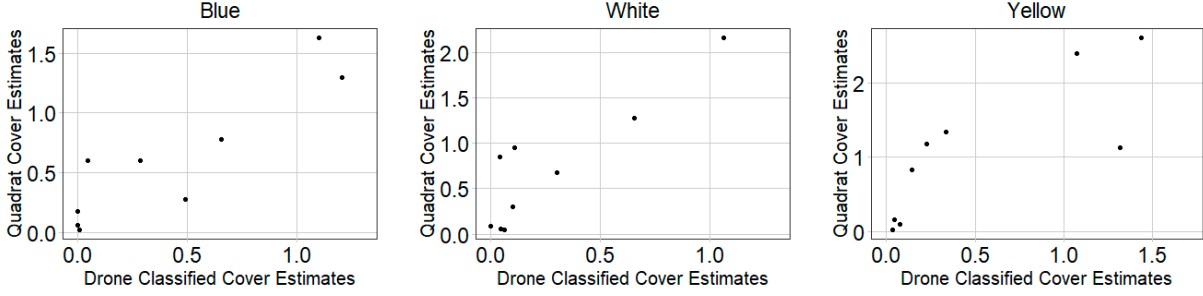

**Figure 4.** Correlation between classified quadrat estimates and ocular quadrat estimates for each color class.

Classified estimates and ocular macroplot estimates were positively correlated between the floral classes but exhibited a weaker relationship than the classified estimates and quadrat estimates (Figure 6). Blue flowers had an r-squared value of 0.41, white flowers had a value of 0.80, and yellow flowers had a value of 0.34.

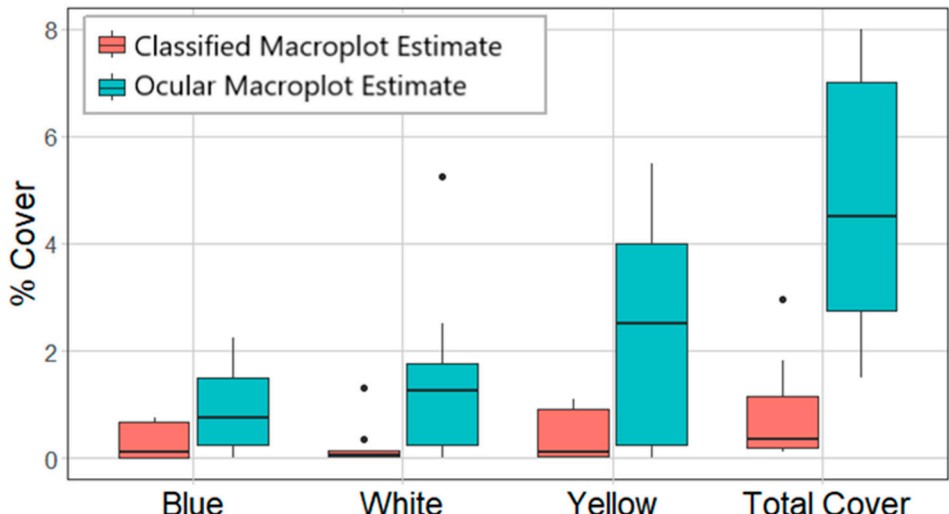

**Figure 5.** Comparison of floral cover between classified macroplot estimates and ocular macroplot estimates.

**Table 7.** Summary statistical results from the Wilcoxon Rank Sum Test when comparing classified macroplot estimates to ocular macroplot estimates.

| Class | Classified Macroplot Average | Ocular Macroplot Average | Sig Diff | V Score | *p* Value |
|---|---|---|---|---|---|
| Blue | 0.28 | 0.94 | Yes | 36 | 0.014 |
| White | 0.23 | 1.53 | Yes | 44 | 0.007 |
| Yellow | 0.39 | 2.33 | Yes | 42 | 0.024 |
| Total | 0.90 | 4.81 | Yes | 45 | 0.004 |

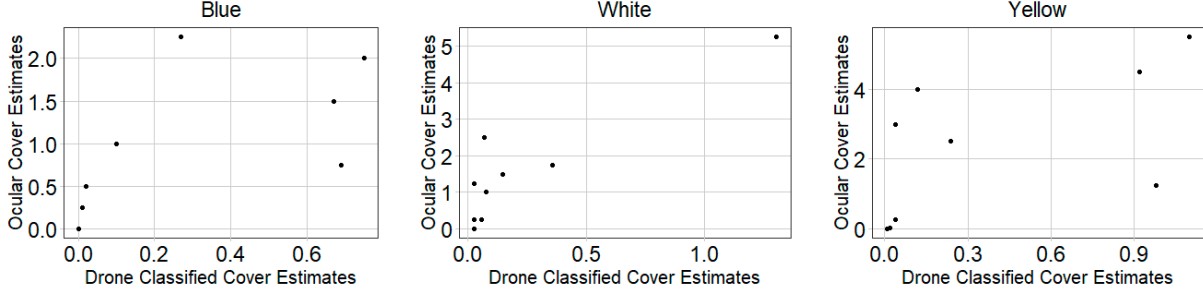

**Figure 6.** Correlation between classified macroplot estimates and ocular macroplot estimates for each color class.

## 4. Discussion

Relatively few studies have applied remote sensing techniques to study insects, insect habitats, and the ecosystem services that they provide [16], including pollinators. Studies that have included insects predominantly used coarse, large-scale landcover data to assess vegetation factors, leaving the finer drivers of pollinator population dynamics within these habitats (floral cover, pollen and nectar availability, or nesting resources) largely unexplored by remote sensing technology [17]. Floral cover and floral richness appear to be broadly associated with pollinator abundance, richness, and visitation rates [18,19]. The use of sUAS systems to assess floral cover has been largely confined to agricultural monocultures in predicting crop yield [20–22]. Horton et al. [23] used sUAS to monitor peach flower blossoms with an average blossom pixel detection rate of 84.3%. Wang et al. [22] used a machine vision assessment on the sUAS imagery of mango orchard blossoms in assessing peak flowering periods and determine the relationship between blossoms and fruit yield. Understanding phenology in an agricultural setting can be useful to predict yield

and timing and potentially assess the stocking requirements for domesticated pollinators. However, few studies have attempted to investigate phenological cycles in heterogeneous environments with a high floral diversity due to spectral complexity [24] and spectral variability at fine scales. Barnsley et al. [15] conducted a study concurrently with this study and explored the use of hyperspectral sUAS imagery to assess spatiotemporal floral resource dynamics with high accuracy. Likewise, Gallmann et al. [14] utilized high-resolution drone imagery and deep learning to assess floral counts compared to manual count methods. A recent call for more spatial data describing the temporal-spatial distribution of flowers was issued by Gonzalea et al. [4], who emphasized the potential utility of sUAS in fine-scale spatiotemporal floristic surveys.

The supervised classification of high-resolution sUAS-generated orthomosaics in this study reflected general ground cover and floral cover in subalpine meadow systems with high accuracy—roughly 74 to 100% (Table 4). Variations in the accuracy between floral cover classes (blue, white, and yellow) occurred for a variety of potential reasons. For instance, the pixels that formed the gradient between shadowed vegetation and brighter green vegetation often possessed a very similar spectral signature to blue flowers at the study sites, indicating the importance of mid-day data collection when shadows are minimal. Likewise, white limestone rocks scattered throughout the sites often possess similar spectral signatures to those of white flowers. Yellow flowers, however, had no such elements within the dataset that shared a similar spectral signature and, therefore, resulted in the highest user and producer accuracies.

The success of these sUAS floral monitoring missions further supports the findings and conclusions of other recent and related studies [4,14,15]. Due to the intra- and interspecific spectral overlap and variation within each color class, individual species within classes are difficult to differentiate using this rapid classification technique. However, the broader color classes and overall floral cover can still be valuable metrics. Understanding these broader cover types and their quantities across the landscape may correlate with individual species or functional insect groups that are utilizing specific resources within the area.

While utilizing drones can help land managers assess floral and vegetative cover throughout and across seasons, accurate cover values alone cannot indicate the pollen and nectar availability held within the blooms. A proximal step to understanding existing resource availability for honeybees in a given area could incorporate the double sampling of nectar and pollen loads by species or color class cover. The metrics of honeybee resource consumption are known [25]. If nectar and pollen availability show a consistent correlation to floral cover, their availability in a given area can be estimated, and appropriate stocking limits applied using such a method.

In addition to the evaluation of supervised classification accuracy, this study provides a side-by-side comparison of two traditional ocular methods for measuring floral cover (quadrat and macroplot). By observing average cover estimates and correlations between the traditional ocular methods and sUAS-classified imagery, we describe some of the advantages and potential pitfalls of these traditional methods in contrast to a direct comparison of sUAS-classified imagery taken of the same areas.

The ocular quadrat estimates seem to reflect similar results to their corresponding classified estimates than the ocular macroplot method, though the difference between the two resulted in a significant overestimation across all floral color classes by roughly 0.7 to two times ($p$-values ranging from 0.055 to 0.0117). Meter-squared quadrats allow the technician to look directly over the entire area being measured, granting the same perspective as UAS-capturing aerial imagery. This method offers the advantage of directly observing each individual species within the quadrat as well as their generalized floral class.

The most rapid assessment of floral cover, the ocular macroplot, was the least consistent and strayed furthest from the accurately classified floral metrics, overestimating by roughly three to seven times depending on the floral class. The cover assessment of each individual, floral class, as well as the overall floral cover, was statistically significant

between ocular and classified methods for the macroplot (*p*-values ranging between 0.014 and 0.004). This could be due to the horizontal parallax encountered when assessing floral cover across a vast area from the ground. Radoux et al. [26] found that even the residual parallax error from satellite imagery taken at an angle positively biased toward forest cover metrics concluded a need for near-nadir images to accurately map landscapes. From the ground perspective, the human eye cannot perceive the interstitial spacing and low-lying cover between plants as the sUAS can from its vertical perspective. Furthermore, the horizontal field of view may extend the perception of floral cover when looking across plants instead of directly down on them. In addition to perspective bias, observer bias based on the complexity of the ecosystem and the field technician's experience and training may result in a greater or lower estimated accuracy [27].

DiMaggio et al. [28] used sUAV double sampling techniques in arid rangelands to estimate forage availability for cattle. Likewise, in this study, a strong correlation between ground and aerial floral cover estimations could allow for a more accurate assessment of pollinator resource availability. Though ocular quadrat estimations differed significantly in scale from their classified counterparts, cover values were highly correlated with one another across all floral classes (r-squared values ranging from 0.66 to 0.81). This correlation suggests that traditional quadrat estimation can accurately assess the trends and relative cover between classes but may fail to accurately describe floral cover as a percentage of overall ground cover with the same accuracy as the sUAS. The accuracy of traditional quadrat methods like the marcoplot methods may rely on technician experience and spatial assessment capability more heavily than methods utilizing sUAS imagery.

The strong correlation between quadrat and classified estimates, in conjunction with the high accuracy of classified images, indicates that while an individual's perception may vary in scale from the true cover value, the perception of relative proportion remains consistent. This suggests that by using a double sampling protocol, an individual can calibrate and correct quadrat measurements taken on the ground to increase their accuracy. This method could help reduce inter-technician bias over long-term or large-scale projects.

Ocular macroplot estimates did not correlate well with all classified floral classes. Blue and yellow floral classes correlated poorly at 0.41 and 0.34, respectively, while the white floral class had a relatively high correlation at 0.80. These findings suggest that ocular macroplots may not be a reliable method for quantifying floral cover across large areas and may only be useful for assessing the relative differences between color classes.

While there are more advanced sensors and classification methods for image classification, the simplicity of this method and drone platform lends itself to less technically trained land managers, graduate students, or agency personnel who are seeking to utilize this technology. It also provides an avenue that is much less cost-prohibitive than other, more technical alternatives.

This methodology is not only highly accurate but also consistent across flights. Many research studies utilize a team of plant surveyors, each with their own perception of cover, despite tools and aids to help measure them more accurately. The varying perception of cover between technicians can introduce inconsistencies and errors, especially across long-term datasets with revolving technicians and personnel. sUAS-derived imagery not only produces consistent and accurate classifications but also provides a census record of the areas that are flown over. These records can be reviewed and validated for years afterward both visually and quantitatively.

sUAS missions are simple to construct and are adaptable in the field. sUAS can be launched from accessible roadways and rapidly flown up steep slopes, across drainages, or over otherwise difficult terrain. Vegetative metrics collected using sUASs save field time and cover more ground in the same amount of time as traditional methods. Instead of a team of technicians, surveys can be performed by a single individual. This simple supervised classification approach requires little back-end post-processing and can be performed by a trained technician in less than a day. Furthermore, these data can be

post-processed at any time of the year, allowing for more thorough investigations of a site outside of the field's season.

Both the accessibility and the sophistication of sUAS platforms are rapidly expanding. Since the initiation of this study, drones have become smaller, sensors have become more powerful, and battery life has increased. Likewise, the capability of back-end post-processing platforms, automation, and machine learning techniques have also evolved. As this technology improves, it can continue to present novel approaches and applications in the realm of natural resource management. Land managers and scientists alike should strive to learn about and integrate this technology where appropriate into their projects and fieldwork.

## 5. Conclusions

Floral assessments and surveys are becoming increasingly important as the interest in wild pollinator and honeybee habitats, and their health becomes a focal point for public land managers. The use of sUAS in floral cover surveys can be quickly and accurately executed and replicated throughout and across seasons to monitor these resources. Equivalent census sampling performed on the ground requires immensely greater efforts in both time and crew size. When compared to traditional field methods, sUAS-derived imagery and classification differed significantly in their cover estimation for all floral classes; however, general trends were reflected using the quadrat method. Ocular macroplots appeared to be inconsistent in both estimates and trends when compared to the sUAS method. Individual species identification was limited with the techniques employed in this study, indicating that this method may be best utilized initially in tandem with some on-the-ground surveys to establish species composition data. With the rapid development of sUAS sensors, it is important for land managers to incorporate the use of these tools so they can increase efficiency and consistency and standardize estimates within their management regime.

**Author Contributions:** Conceptualization, N.V.A., V.J.A. and S.L.P.; methodology, N.V.A. and V.J.A.; software, N.V.A. and S.L.P.; validation, S.L.P., N.V.A. and V.J.A.; formal analysis, N.V.A. and T.J.T.; investigation, N.V.A.; resources, S.L.P., R.L.J. and V.J.A.; data curation, N.V.A.; writing—original draft preparation, N.V.A.; writing—review and editing, N.V.A., S.L.P., V.J.A. and R.L.J.; visualization, N.V.A. and T.J.T.; supervision, S.L.P. and V.J.A.; project administration, N.V.A., S.L.P. and V.J.A.; funding acquisition, V.J.A. and R.L.J. All authors have read and agreed to the published version of the manuscript.

**Funding:** This research was funded by The United States Forest Service, grant number 16-CS-11041000-022. Funding was supplemented internally by Brigham Young University.

**Data Availability Statement:** The data presented in this study are publicly available on Figshare and can be found at https://figshare.com/s/e3532425dc535d5e9f23. doi: 10.6084/m9.figshare.24796986 (accessed on 15 October 2023).

**Conflicts of Interest:** The authors declare no conflicts of interest. The funders had no role in the design of the study; in the collection, analyses, or interpretation of data; in the writing of the manuscript; or in the decision to publish the results.

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
