# Peer review of "Detecting Floral Resource Availability Using Small Unmanned Aircraft Systems"

_land, doi:10.3390/land13010099_

Round 1

Reviewer 1 Report

Comments and Suggestions for Authors

Dear authors!

The article "Detecting Floral Resource Availability Using Small Unmanned Aircraft Systems" is an interesting scientific study aimed at comparing methods for assessing the availability of floral resources using large-scale remote sensing. The article is well structured and provides an extensive literature review on the research problem.

Sections of the research results and discussion are sufficiently fully disclosed for understanding the research conducted.

I believe that this article can be published in its present form.

Author Response

Dear Reviewer,

Thank you for your time and thorough review of our work. I believe the feedback to publish “as is” reflects the hard work and diligence our research team has put in to present a quality paper for review. Upon this review we did add some further elements and details to the methods and discussion sections which we hope will improve the understanding and relevance of our work.

Thank you again for your thoughtful review and constructive comments.

Reviewer 2 Report

Comments and Suggestions for Authors

Dear corresponding Author,

the aim of  paper reviewed is to illustrate a methodology by Small Unmanned Aircraft Systems (sUAS) acquisition useful for accurate broad floristic surveys and an additional solution to more traditional alternative methods of vegetation assessment.  The methodology explained in the paper was designed as a simplified approach using tools frequently available to land managers. The objectives proposed by this study were assess the efficacy of using high resolution imagery from sUAS to accurately calculate floral cover. and compare the results of the classified aerial imagery to two traditional methods of vegetation cover analysis. The study utilizes a standard off the shelf drone platform and RGB sensor to assess floral cover providing an affordable, simplistic, and rapid solution for technicians and land managers currently utilizing traditional methods less efficient.

Overall, the study touches a topical subject in the context of remote and or proximal sensing imagery classification in precision agriculture . My general opinion of results showed in the paper is that there are some issues to be addressed before the manuscript can be suitable for publication. I suggest improving the text following my comments and suggestions in the PDF file attached.

My review response of this paper is minor revision.

Good luck!

ZAP

Author Response

Dear Reviewer,

Thank you for your time and thorough review of our work. The comments and feedback you have given us has helped us refine and enhance this article for which we are very appreciative. The specific edits you provided in your PDF helped us understand what items specifically we needed to clarify and are each addressed below:

Line 35 We included the mention of land cover survey.

Line 41 We rewrote to the sentence to further clarify.

Line 49 I agree that spectral is a better suited word to digital.

Line 51 We clarified why satellites are not a good solution for monitoring small and variable seasonal resources.

Line 83 While we did classify some broader vegetation groups, the focus of the study is on floral resources specifically within the context of pollinator resources, so we would like to keep it as is.

Line 85 We thought the contrast was appropriate as the simplistic remote sensing platform we used is quite different from the sensors used in the other two studies.

Line 103 National land cover and regional land cover classes typically represent different community types or land uses. These flights were all within the same subalpine community type, our classes were selected to help us extract and calculate the floral cover at each site. Justification for the classes we used has been added.

Thank you again for your thoughtful review and constructive comments.

Reviewer 3 Report

Comments and Suggestions for Authors

This study discusses a simplified drone technology for the identification of grassland flowers. This technology holds significant potential for applications, and the quality of the paper's writing is commendable. It is recommended for publication with minor revisions.

 In the methods section, it is advisable to provide a more detailed description, including specific details about the supervised methods and parameters.

In the figures and tables, it is suggested to round the data to two decimal places.

It is recommended to thoroughly explain the reasons for the significant differences in classification accuracy among different flower colors.

The authors can use "%" instead of "134-140 percent."

Some land managers may find it challenging to use specialized software such as ENVI. Therefore, the authors might consider developing related programs for integration with professional software to enhance the user-friendliness of the identification technology.

Author Response

Dear Reviewer,

Thank you for your time and thorough review of our work. The comments and feedback you have given us has helped us refine and enhance this article for which we are very appreciative. Each comment is addressed below:

Comment 1: Some additional details have been provided. Our methodology, however, was simplistic by design therefore there is not very much additional relevant information to share regarding our classification techniques.

Comment 2: The tables look much cleaner with just two decimal places, good catch! We adjusted all data to two decimal places apart from whole numbers and p-values.

Comment 3: This was certainly an oversight as there were some clear reasons throughout the classification process as to why some color classes were more accurately classified than others. This has been added to the discussion section.

Comment 4: We changed percent to %

Comment 5: Great idea for subsequent research and development, imagery programs are often difficult for lay people an the development of software programs that are more simplistic to use for land managers would be beneficial. This exceeds the scope and budget of this particular project but something to look toward in the future.

Thank you again for your thoughtful review and constructive comments.